# Effect of Two-Step Sintering on the Mechanical and Electrical Properties of 5YSZ and 8YSZ Ceramics

**DOI:** 10.3390/ma16052019

**Published:** 2023-02-28

**Authors:** Yunpeng Li, Hongqian Sun, Jing Song, Zhiyu Zhang, Hao Lan, Liangliang Tian, Keqiang Xie

**Affiliations:** 1National Engineering Research Center for Green Recycling of Strategic Metal Resource, Institute of Process Engineering, Chinese Academy of Sciences, Beijing 100190, China; 2College of Metallurgy and Energy Engineering, Kunming University of Science and Technology, Kunming 650093, China; 3Ganjiang Innovation Research Institute, Chinese Academy of Sciences, Ganzhou 341119, China; 4School of Electronic Information and Electrical Engineering, Chongqing University of Arts and Sciences, Chongqing 402160, China

**Keywords:** YSZ, two-step sintering, grain refining, fracture toughness, electrical conductivity

## Abstract

Yttria-stabilized zirconia (YSZ) has been widely used in structural and functional ceramics because of its excellent physicochemical properties. In this paper, the density, average gain size, phase structure, and mechanical and electrical properties of conventionally sintered (CS) and two-step sintered (TSS) 5YSZ and 8YSZ are investigated in detail. As the grain size of YSZ ceramics became smaller, dense YSZ materials with a submicron grain size and low sintering temperature were optimized in terms of their mechanical and electrical properties. 5YSZ and 8YSZ in the TSS process significantly improved the plasticity, toughness, and electrical conductivity of the samples and significantly suppressed the rapid grain growth. The experimental results showed that the hardness of the samples was mainly affected by the volume density, that the maximum fracture toughness of 5YSZ increased from 3.514 MPa·m^1/2^ to 4.034 MPa·m^1/2^ in the TSS process, an increase of 14.8%, and that the maximum fracture toughness of 8YSZ increased from 1.491 MPa·m^1/2^ to 2.126 MPa·m^1/2^, an increase of 42.58%. The maximum total conductivity of the 5YSZ and 8YSZ samples under 680 °C increased from 3.52 × 10^−3^ S/cm and 6.09 × 10^−3^ S/cm to 4.52 × 10^−3^ S/cm and 7.87 × 10^−3^ S/cm, an increase of 28.41% and 29.22%, respectively.

## 1. Introduction

YSZ is widely used as a structural and functional ceramic because of its excellent mechanical properties [1], electrical properties, and thermal stability. Among them, 3 mol% yttria-stabilized zirconia (3YSZ) usually exists as a tetragonal phase with excellent mechanical properties and is mainly used in sealing devices, filter materials, and dental materials, etc. 8YSZ usually exists as a cubic phase [2] and has good ionic conductivity and piezoelectric effect; moreover, it is mainly used in solid fuel cells, oxygen sensors, and piezoelectric ceramics. The mechanical and electrical properties of ceramics are closely related to their densification [3]. Improving the mechanical or electrical properties of densified ceramics usually starts with powder processes such as finer particle size distribution [4,5], doped second phases [6,7,8,9], or suitable sintering processes such as hot press sintering [10], microwave sintering [11], spark plasma sintering [12,13], and two-step sintering (TSS) [14,15,16,17,18,19,20,21], etc. Compared with conventionally sintered (CS) compacts, a reduction in density difference in the early stage of TSS leads to a decrease in the spatial scale of density fluctuation. The smoother pore channel results in a delay in the extrusion of the pore channel to high density, resulting in a smaller average grain size and a more uniform grain size distribution [14]. The TSS was first proposed by Chu et al. in the 1990s [15], and an improved TSS process was proposed by Chen and Wang [16]. This process has become the mainstream TSS process, and is based on the principle that the activation energy of grain growth is lower than the activation energy of densification [17], where the sample is heated to a high-temperature T1 and then rapidly cooled to a low-temperature T2 for a long holding time. Using this method, the grain boundary diffusion of the sample can be maintained and grain boundary migration can be controlled, which further inhibits grain growth. The TSS sintering of YSZ ceramics at low temperatures not only reduces sintering costs and production cycle time but also prevents microstructural coarsening of YSZ and undesirable reactions between YSZ and other materials. The grain refinement resulting from this method of TSS not only helps to improve mechanical properties [18,19] but also improves electrical conductivity [20,21]. The mechanical and electrical properties of 3YSZ and 8YSZ in TSS have been reported; however, less has been reported about the properties of 5YSZ. In the existing reports, the sintering time of the second part is often too long, and the phase structure and conductivity activation energy are not described in much detail.

The differences in the mechanical and electrical properties of two high-performance ceramics (5YSZ/8YSZ), CS and TSS, were compared in detail in this study. Unlike the previous scholars’ studies on TSS, the authors did not use a heat preservation time that is too long after T2. This provides a reference for the two-step sintering of 5YSZ in a short time, and the structure and properties of this process were studied in detail. In this study, a preliminary study on the mean grain size and pore size evolution was first conducted to determine the appropriate heating temperatures T1 and T2. Then, the effects of TSS on the structure and properties of the 5YSZ and 8YSZ ceramics were discussed by applying the suitable temperatures T1 and T2. 5YSZ and 8YSZ ceramics have good ionic conductivity, therefore they can be used as solid electrolytes in automobile exhaust sensors and fuel cells.

## 2. Experiment and Test Methods

### 2.1. Experiment

High-purity 5YSZ and 8YSZ prepared in the laboratory were used as raw materials. Using a dry press HY-15 (Keqi, Tianjin, China), a certain amount of powder was weighed and cold pressed axially in a steel cylindrical mold with an inner diameter of 10 mm and pressed at 4 MPa, followed by sintering. The temperature range of CS was 1300–1550 °C with a heating rate of 3 °C/min and holding at the highest temperature for 2 h. The process of TSS was firstly raised to T1 at a heating rate of 3 °C/min without holding and cooled to a lower temperature T2 at 10 °C/min, holding for 2 h, 5 h, and 10 h, respectively, and then cooling with the furnace.

### 2.2. Test Methods

The particle size of the powders was studied using a laser particle size analyzer Mastersizer 2000 (Marvin, Los Angeles, CA, USA). A scanning electron microscope JSM-7610F (JEOL, Tokyo, Japan) and a BET-specific surface fully automated physisorption instrument ASAP 2020HD88 (Micromeritics, Atlanta, GA, USA) was used to determine the morphology of the ground ceramics and the surface area of the powders, respectively.

The volume density of the ceramic samples was measured using Archimedes’ drainage method. After the samples were polished and gold sprayed, the surface morphology was observed using scanning electron microscopy. The average grain size was calculated by counting no less than 100 grains using Nano Measurer 1.2 software. Tests in the range of 5–90° (Cu-Kα_1_ radiation, λ = 0.15406 nm) were performed in an X-ray diffractometer Empyrean (PANalytical B.V., Almelo, The Netherlands). The Raman analyzer was used to confirm the presence of both cubic and tetragonal phases. X-ray diffraction (XRD) measurements were also performed in a narrow the 2θ range from 70° to 80° with a step size of 0.05° and 30 s of exposure time per step to calculate the amount of each phase. The calculation formula is shown in Equation (1) [22]:(1)$$\frac{M_{c}}{M_{t}}=0.88\frac{I_{(400)c}}{I_{(400)t}+I_{(004)t}}$$
where *M_c_* and *M_t_* are the mole fractions of the cubic and tetragonal phase over 2θ range of 73–75°, respectively; the combined intensities of the various reflections are expressed by the appropriate Miller indices. The program celref3 was implemented for Rietveld refinement of XRD data to determine cell parameters [23]. The refinement was performed for various combinations of space groups, including cubic (FM-3M) and tetragonal (P42/NMC) before and after the subtraction of the background.

Vickers hardness and fracture toughness were assessed using a Vickers hardness tester TH700 (Shidai, Beijing, China). Indentation tests were performed on polished samples with loads of 49 N for 5YSZ and 8YSZ, holding for 10 s. The hardness Hv was calculated according to ASTM C1327 standard [24], and the diagonal length of the indentation was determined using metallographic microscopy observation with the following Equation (2):(2)$$Hv=\frac{1.854P}{d^{2}}$$
where *P* is the applied load (in N) and *d* is the average diagonal length (in μm). The fracture toughness K_IC_ was determined by measuring the indentation in the following Equation (3) [25,26,27]:(3)$$K_{IC}=0.0016{\left(\frac{E}{Hv}\right)}^{1/2}{\left(\frac{P}{C^{3/2}}\right)}^{1/2}$$
where *E* is Young’s modulus (in GPa *E* was chosen to be 210 GPa) [28], *H_v_* is the test hardness (in GPa), *P* is the load (in N), and c is half of crack length (in mm). For each result, at least five indentations were performed, and the average values are reported. The elastic modulus was measured using a nanoindenter G200 (MTS, Eden Prairie, MN, USA) for five points of a straight line with an indenter resolution of 1 nN and 0.01 nm for load and displacement, respectively.

The two-probe DC technique was used to measure the conductivity of YSZ with different average grain sizes, and the grain conductivity and grain boundary conductivity of YSZ were investigated using impedance spectroscopy. Pt electrodes were coated on the relative surfaces and dried at 1000 °C for 2 h. The samples were then placed in a fixture with a two-electrode configuration in a horizontal tube furnace. The impedance analyzer Vertex.C.EIS (Ivium, PSV Eindhoven, The Netherlands) was used for the measurement of the samples in an atmosphere-simulating air environment. The measurement temperatures were 350 °C, 450 °C, 550 °C, 680 °C, and 800 °C. The conductivity is calculated as in Equation (4):(4)$$\sigma =\frac{L}{R\times S}$$
where *L* is the thickness of the sample (in cm) and *S* is the electrode area (cm^2^).

## 3. Results and Discussion

### 3.1. Analysis of YSZ Powders

The particle size distribution of the homemade YSZ powder after grinding is shown in Figure 1a. 5YSZ and 8YSZ powders have similar particle size distribution, with d50 between 2–3 μm, and the specific surface area is 8.882 m^2^/g for 5YSZ and 9.854 m^2^/g for 8YSZ. Figure 1b shows the XRD plots of both powders, and it can be seen that both powders have no monoclinic phase; they are both tetragonal and cubic phases.

### 3.2. Volume Density and Average Grain Size of Ceramics

The trends of volume density and the average grain size of the samples in the CS process are shown in Figure 2. For the 5YSZ ceramics, the volume density and average grain size of the samples became larger with an increasing temperature in the sintering range of 1350–1550 °C. When T = 1450 °C, the volume density of the 5YSZ ceramics was 5.961 g/cm^3^, and the average grain size was about 500 nm, at which time the pores had completely disappeared, as shown in Figure 3b. We consider 1450 °C as the temperature at which the 5YSZ ceramics begin densification. This is usually explained by the solid-state sintering theory, whereby the dispersed pores have the function of “pinning” the grain boundaries and preventing grain boundary migration. At the beginning of sintering, there is no “collapse” in the open pores, therefore no closed pores are formed and grain growth is inhibited; as the temperature increases, a large number of closed pores are formed in the middle and late stages of sintering, and the “pinning” effect is strongly weakened and grain growth is promoted. For 8YSZ ceramics, in the sintering range of 1300–1500 °C, the average grain size becomes significantly larger with an increase in temperature, and the density of the sample decreases at T = 1500 °C, which may be caused by the abnormal growth of grains. When T = 1400 °C, the volume density of 8YSZ ceramics is 5.958 g/cm^3^, the average grain size is about 1680 nm, and the pores have completely disappeared by then. The grain growth difference between the two kinds of ceramics during sintering can be reasonably explained by the solute drag mechanism of Y^3+^ ion separation along the grain boundary. The segregation of the Y^3+^ ions directly affects grain growth and is closely related to the driving force of grain boundary segregation-induced phase transformation (GBSIPT) [29].

Table 1 shows the density and average grain size of samples under different TSS systems. With an increase in the sintering temperature and holding time, the samples gradually become dense and the grains grow slowly. Figure 3 shows the morphology of samples under different sintering systems under the holding time of 2 h. The degree of densification of the TSS system depends on T1 and T2, and the average grain size is mainly affected by T2. The TSS samples have a higher density and smaller average grain size at a lower temperature. For the 5YSZ ceramics, when T1 = 1500 °C, T2 = 1400 °C, and t = 2 h, the volume density of the sample is 5.962 g/cm^3^, and the grain size is 400 nm. When T1 = 1400 °C, T2 = 1350 °C, and t = 10 h, the volume density of the sample is 5.945 g/cm^3^, and the grain size is 280 nm. For 8YSZ ceramics, when T1 = 1450 °C, T2 = 1350 °C, and t = 2 h, the volume density of the sample is 5.906 g/cm^3^, and the grain size is 960 nm. When T1 = 1350 °C, T2 = 1300 °C, and t = 10 h, the volume density of the sample is 5.901 g/cm^3^, and the grain size is 410 nm. Compared with CS, TSS can achieve densification at lower temperatures and inhibit grain growth. Its sintering mechanism can be explained using Figure 4, where v is the grain boundary migration rate and T is the thermodynamic temperature. The sintering theory of ceramics [30,31] posits that there are six main sintering mechanisms in the sintering process of powder, and that various sintering mechanisms play a certain role in the sintering process. For traditional sintering, the surface diffusion mechanism plays a leading role at the initial stage of sintering (especially when the relative density of the body is low); however, the grain boundary diffusion is the main sintering mechanism at the end of sintering [32]. Without inhibiting diffusion, as the temperature decreases, the competition between coarsening and densification tends to be beneficial to densification. When the second-step sintering process is carried out at the appropriate temperature (T2), its main mechanism is grain boundary diffusion and bulk diffusion [30].

**Table 1 materials-16-02019-t001:** Sintered densities and grain size of 5YSZ and 8YSZ ceramics under various heating parameters of two-step sintering.

	T1 (°C)	T2 (°C)	t (h)	ρ_v_ (g/cm^3^)	ρ_r_ (%)	G (nm)		T1 (°C)	T2 (°C)	t (h)	ρ_v_ (g/cm^3^)	ρ_r_ (%)	G (nm)
5YSZ	1500	1400	2	5.962	98.83	400	8YSZ	1450	1350	2	5.906	98.60	960
			5	5.981	99.14	410				5	5.924	98.90	1310
			10	6.012	99.66	510				10	5.936	99.10	1480
	1450	1400	2	5.946	98.56	370		1400	1350	2	5.804	96.89	860
			5	5.969	98.95	390				5	5.874	98.06	1010
			10	5.987	99.24	480				10	5.921	98.85	1060
	1450	1350	2	5.891	97.65	270		1400	1300	2	5.791	96.67	470
			5	5.937	98.42	310				5	5.862	97.86	510
			10	5.968	98.93	335				10	5.919	98.81	580
	1400	1350	2	5.846	96.91	200		1350	1300	2	5.622	93.86	280
			5	5.891	97.65	250				5	5.776	96.43	370
			10	5.945	98.55	280				10	5.901	98.51	410

Note: t: holding time, T1: a high temperature, T2: a low temperature, ρ_v_: volume density, ρ_r_: relative density, G: grain size. The relative density of 5YSZ is calculated according to 6.0326 g/cm^3^ theoretical density and 8YSZ is according to 5.99 g/cm^3^ theoretical density [33].

### 3.3. Physical Phase Structure

Figure 5a shows the XRD patterns of four fully dense YSZ ceramics obtained using the CS and TSS processes. All samples contain tetragonal and cubic phases and no monoclinic phase. Comparing the 5YSZ ceramic samples at CS-1500 °C and TSS-1500-1400 °C, the diffraction peaks are stronger for the CS process samples; moreover, for the 8YSZ samples, the characteristic peak (220) peak intensity is higher in the CS process, at CS-1450 °C and TSS-1450-1350 °C. Figure 5b shows the Raman spectra of the two ceramics. A certain amount of tetragonal phase still exists in the two ceramics, which may be related to the holding time. In the TSS process, the tetragonal characteristic peak at 262 cm^−1^ [35] disappears, and there is no significant difference between the peak intensity and peak position at 480 cm^−1^ and 635 cm^−1^ [36], which indicates that there is no excessive lattice distortion during sintering. Table 2 shows the phase ratio and crystal cell parameters of these four kinds of ceramics. In the TSS process, the cubic phase content of 5YSZ ceramics increases greatly, and the phase structure of 8YSZ ceramics has no significant difference. Compared with the CS process, the two ceramics show smaller cell parameters and cell volume in the TSS process. Supporting the assumption of lattice deformation, a decrease in unit cell volume and an increase in its tetragonality level were indicated [37].

### 3.4. Mechanical Properties

Figure 6 shows the trends of hardness and fracture toughness of the two ceramic samples in CS process at different sintering temperatures. The fracture toughness of the 5YSZ and 8YSZ ceramic samples increased and then decreased with the increase in sintering temperature, and reached the maximum value at 1400 °C. The maximum hardness of 5YSZ ceramic samples in the CS process was 15.709 GPa, and the maximum fracture toughness was 3.514 MPa·m^1/2^, while the maximum hardness of 8YSZ ceramic samples was 14.972 GPa and the maximum fracture toughness was 1.491 MPa·m^1/2^. Table 3 shows the hardness, toughness, and elastic modulus values of the ceramic samples in the TSS process. The maximum fracture toughness increased to 4.034 MPa·m^1/2^ and 2.126 MPa·m^1/2^. TSS sintering improved the fracture toughness of the samples, which was mainly due to the grain refinement effect brought about by TSS. The finer the grain size, the larger the grain boundary area, and the more curved the grain boundary is, the more unfavorable to the crack extension, and the stronger the resistance to cracking. The hardness of the two ceramic samples in the TSS process is slightly lower than that in CS process because the relative density is slightly lower.

The annular nanoindentation leads to quasi-plastic deformation, which results in localized aggregation of zirconia crystals with micro-cracks and large cracks along the indentation edges. Figure 7 shows the loads and deflections of the four ceramic samples. The displacements of the four ceramics into the elastic–plastic phase are almost the same, but the TSS process sample has a larger deflection in the plastic phase. The modulus of the elasticity of the 5YSZ and 8YSZ ceramic samples in the CS process is 223.545 GPa and 218.602 GPa, respectively, while the modulus of elasticity of the 5YSZ and 8YSZ ceramic samples in the TSS process is 238.075 GPa and 233.408 GPa, respectively. The average grain size determines the deformation zone around the residual indentation. The decrease in the average grain size is associated with a significant decrease in contact hardness, Young’s modulus, plasticity, elastic deformation fraction and cracking caused by machining, and higher elastic and plastic displacements than that in CS. The results indicate that grain refinement is beneficial to improve the elastic deformation capacity of the material.

### 3.5. Electrical Properties

Impedance spectra usually separate grains, grain boundaries, and electrodes in polycrystalline systems [38]. Figure 8 shows the Nyquist impedance plots for 8YSZ-1450 °C ceramic at different temperatures, with the appearance of two depressed arcs that were attributed to the contribution of grains and grain boundaries. Above 450 °C, the high-frequency grain arcs gradually disappear, and only the grain boundary arcs and low-frequency tails are present, which is due to the electrode surface favoring electron transfer and the high electron activity of the electrode material. As the test temperature increases, the arc moves to lower frequencies. With the growth increase in the test temperature, the grain boundary barrier is broken. At this time, the resistivity is mainly determined by the grain resistivity, and the grain boundary arc disappears, leaving only a semicircle arch.. The equivalent circuit used is [CR][CR][CR] [39], which is a good fit.

Figure 9 and Figure 10 show the total conductivity, grain conductivity, and grain boundary conductivity of the different samples at different test temperatures. The sample code, total conductivity, and calculated activation energy for oxygen ion conduction are shown in Table 4. Moreover, logσT shows an excellent linear relationship with 1000/T. The total conductivity of ceramics obtained at different sintering temperatures at low temperatures varies greatly and tends to increase with an increase in test temperature. At 680 °C, the total conductivity of 5YSZ-7 is the highest, 4.52 × 10^−3^ S/cm, which increased by 28.41% compared with 5YSZ-1. At the same time, 8YSZ-6 is 7.87 × 10^−3^ S/cm, which increased by 26.12% compared with 8YSZ-2. At 680 °C, the grain boundary resistance is small, and the total resistance is mainly affected by the grain resistance. The grain resistance can be greatly reduced through TSS, thus reducing the grain resistance and improving the total conductivity. The activation energy of the grain conductivity of 5YSZ ceramics in the TSS process is lower than that in the CS process, while the activation energy of grain boundary conductivity is the opposite, and the activation energy of total conductivity is lower in CS process. The change rule of the activation energy of the grain conductivity of 8YSZ ceramics is consistent with that of 5YSZ ceramics; however, the total conductivity and grain boundary conductivity are slightly different.

The ionic conductivity of zirconia is mainly determined by the crystal morphology of the zirconia solid solution. Compared with the tetragonal phase, the cubic phase has a higher concentration of oxygen vacancies. The cubic phase content of 5YSZ ceramics increased by TSS process, but the cubic phase content of 8YSZ ceramics changed little. With a decrease in the average grain size, the thickness between grains decreases, the grain area increases, and the total effective conductive area increases, making the oxygen vacancy migration easier.

## 4. Conclusions

(1) The degree of the densification of the TSS system depends on T1 and T2, and the average grain size is mainly affected by T2. TSS affords the samples a higher density and smaller average grain size at a lower temperature. The main mechanism of TSS is grain boundary diffusion and bulk diffusion;

(2) The hardness of the samples is mainly affected by the relative density, and the fracture toughness and elastic modulus of the ceramic bodies are increased due to the grain refinement effect;

(3) The total conductivity of the samples is mainly influenced by the grain size and grain boundary area of the electrolyte. 5YSZ and 8YSZ samples increased the maximum total conductivity from 3.39 × 10^−3^ S/cm and 4.52 × 10^−3^ S/cm to 6.09 × 10^−3^ S/cm and 7.87 × 10^−3^ S/cm under 680 °C.

## Figures and Tables

**Figure 1 materials-16-02019-f001:**
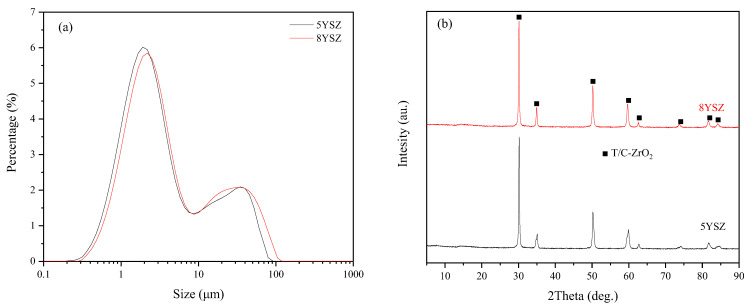
(**a**) Particle size distribution diagram (**b**) XRD diagram of two YSZ powders.

**Figure 2 materials-16-02019-f002:**
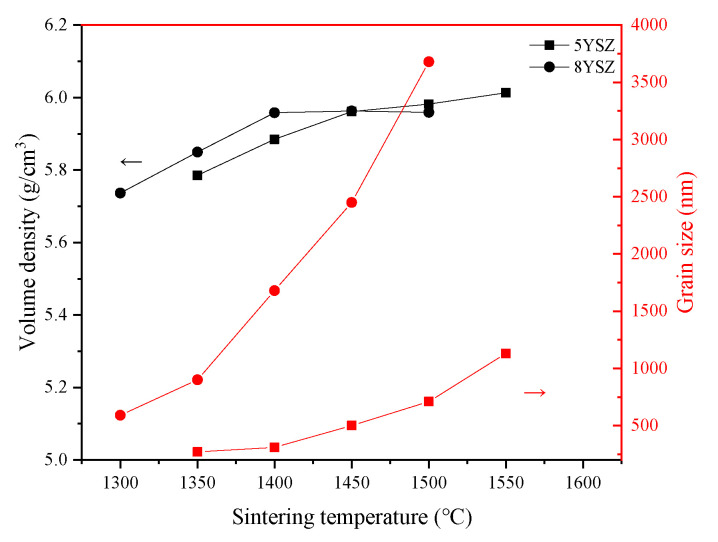
Density and grain size sintered in CS process.

**Figure 3 materials-16-02019-f003:**
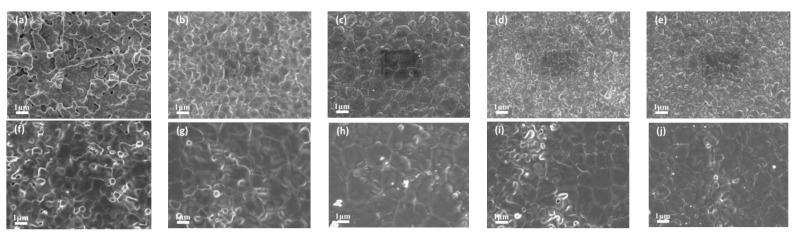
SEM under different sintering modes under 2 h holding time: (**a**) 5YSZ-1400 °C, (**b**) 5YSZ-1450 °C, (**c**) 5YSZ-1500 °C, (**d**) 5YSZ-1450-1400 °C, (**e**) 5YSZ-1500-1400 °C, (**f**) 8YSZ-1350 °C, (**g**) 8YSZ-1400 °C, (**h**) 8YSZ-1450 °C, (**i**) 8YSZ-1400–1350 °C, (**j**) 8YSZ-1450–1350 °C.

**Figure 4 materials-16-02019-f004:**
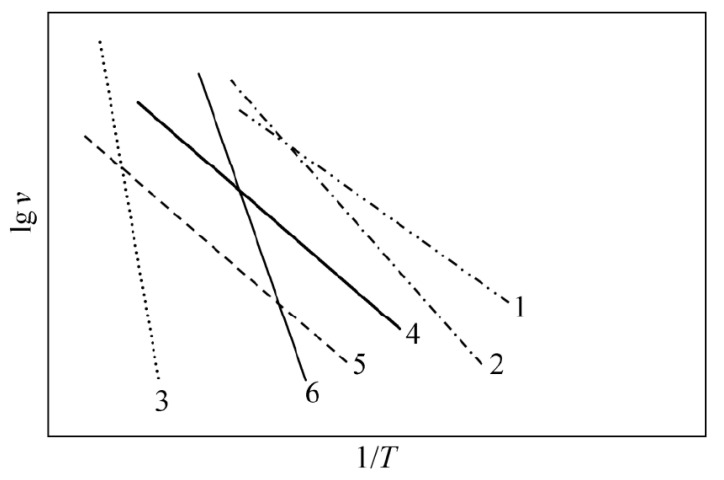
Schematic representation of Arrhenius plot for sintering mechanisms [30,34] (1—Surface diffusion mechanism; 2—body diffusion mechanism; 3—evaporation and accumulation mechanism; 4—grain boundary diffusion mechanism; 5—Grain boundary migration mechanism; 6—Intersection migration mechanism).

**Figure 5 materials-16-02019-f005:**
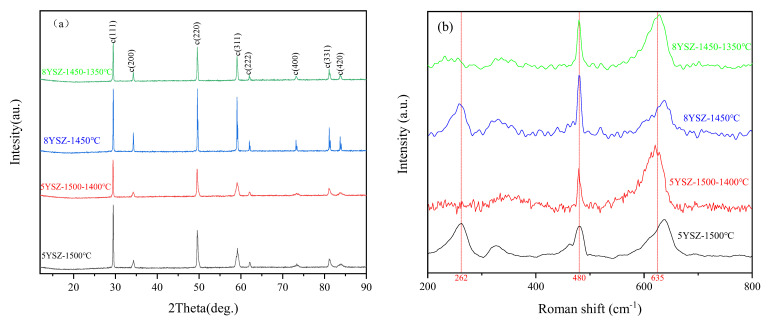
(**a**) XRD patterns; (**b**) Raman patterns of four YSZ samples under 2 holding time.

**Figure 6 materials-16-02019-f006:**
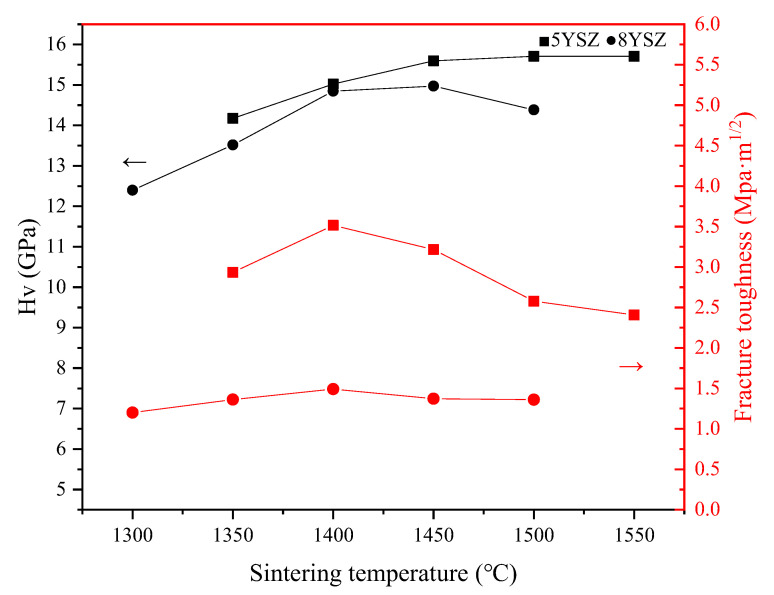
Change trend of hardness and fracture toughness with temperature under CS process.

**Figure 7 materials-16-02019-f007:**
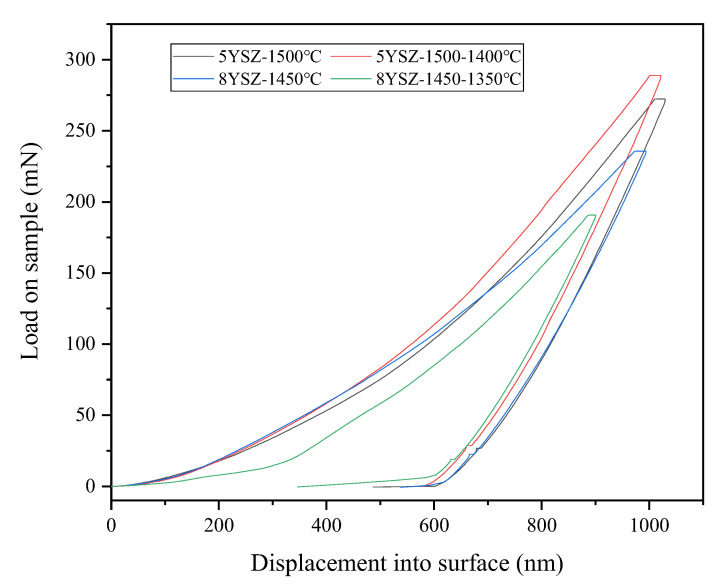
Displacement load diagram of four samples.

**Figure 8 materials-16-02019-f008:**
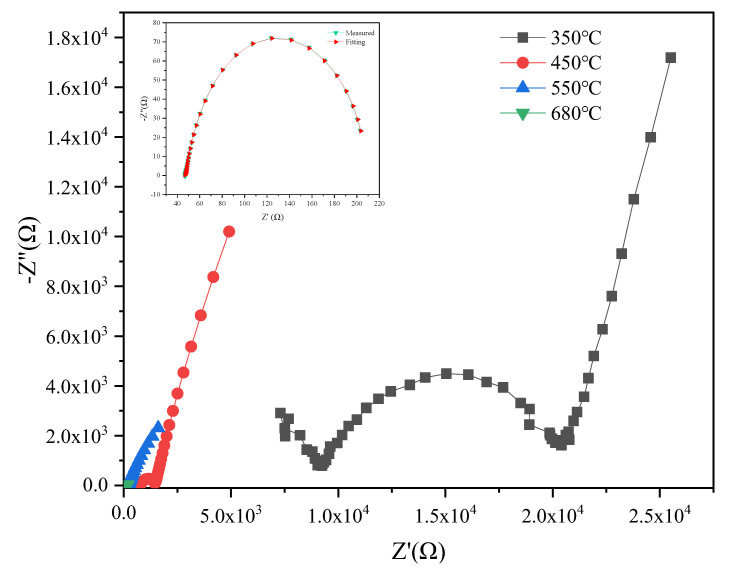
Nyquist diagram under different temperatures with 8YSZ-1450 °C.

**Figure 9 materials-16-02019-f009:**
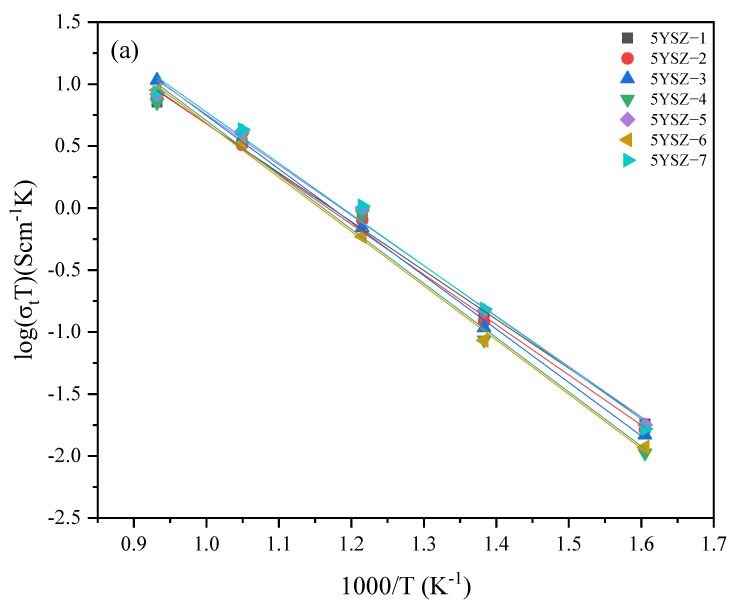
(**a**) Total conductivity, (**b**) grain conductivity, (**c**) grain boundary conductivity of seven 5YSZ ceramics.

**Figure 10 materials-16-02019-f010:**
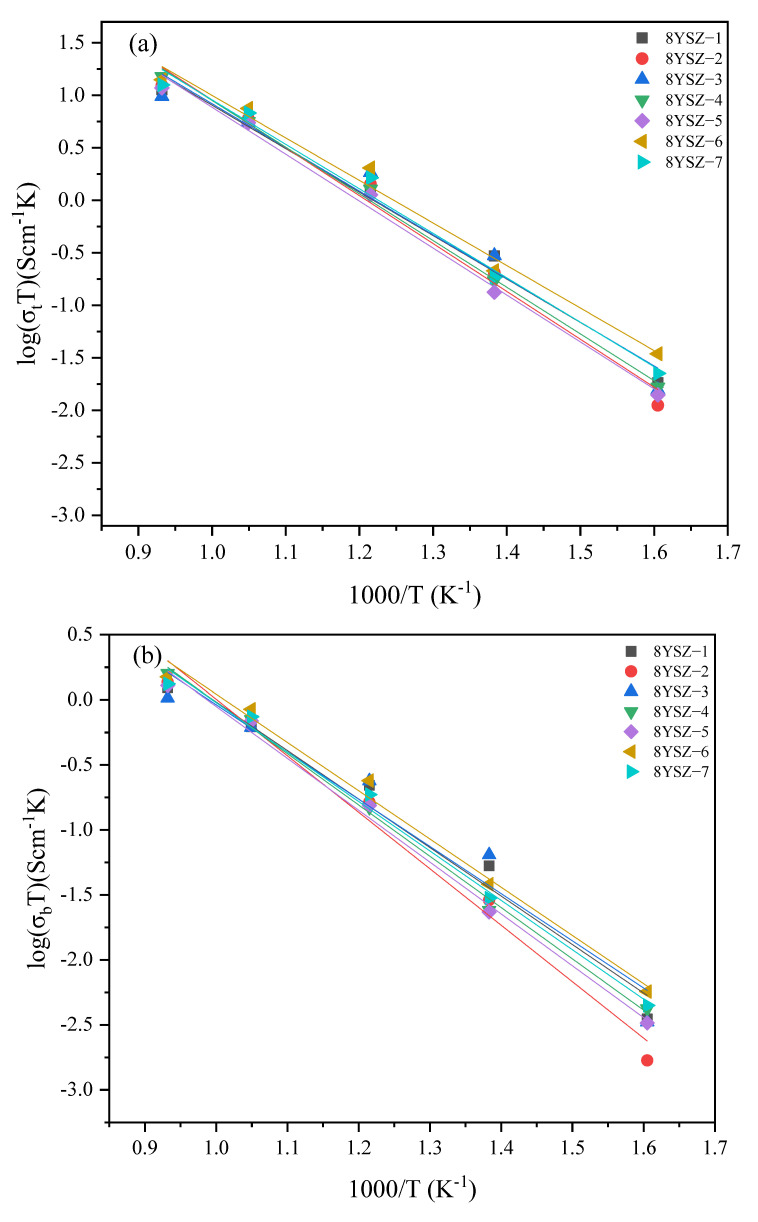
(**a**) Total conductivity, (**b**) grain conductivity, (**c**) grain boundary conductivity of seven 8YSZ ceramics.

**Table 2 materials-16-02019-t002:** The phase composition, cell parameters, and cell volume of four ceramics at different sintering conditions.

Specimen	Phase Composition, %	Latticle Options, Å		c/a	V_cell_, Å^3^
		a	c		
5YSZ-1500 °C	47.52%C	-	5.1705	-	138.23
	52.48%T	3.6589	5.1677	1.413	69.18
5YSZ-1500-1400 °C	59.05%C	-	5.1641	-	137.72
	40.95%T	3.6513	5.1622	1.414	68.82
8YSZ-1450 °C	94%C	-	5.1723	-	138.37
	6%T	3.6644	5.1803	1.414	69.56
8YSZ-1450-1350 °C	94.71%C	-	5.163	-	137.63
	5.29%T	3.6427	5.1732	1.42	68.64

**Table 3 materials-16-02019-t003:** Relationship between mechanical properties, relative density, and grain size of different samples at 2 h holding time.

Sample	Vickers Hardness GPa	Fracture Toughness Mpa·m^1/2^	Elastic Modulus GPa
5YSZ-1500-1400 °C	15.301	3.289	238.075
5YSZ-1450-1400 °C	14.828	4.034	-
5YSZ-1450-1350 °C	14.076	2.797	-
5YSZ-1400-1350 °C	12.861	2.902	-
8YSZ-1450-1350 °C	15.134	1.585	233.408
8YSZ-1400-1350 °C	14.320	2.126	-
8YSZ-1400-1300 °C	13.636	1.817	-
8YSZ-1350-1300 °C	12.507	1.36	-

**Table 4 materials-16-02019-t004:** Activation energy of total conductivity of various samples.

Specimen	Sintering Conditions	σ_t_ (680 °C)	Activation Energy E_a_ (eV)			Specimen	Sintering Conditions	σ_t_ (680 °C)	Activation Energy E_a_ (eV)		
	°C	×10^−3^ S/cm	σ_t_	σ_b_	σ_gb_		°C	×10^−3^ S/cm	σ_t_	σ_b_	σ_gb_
5YSZ-1	1500-2 h	3.52	0.72	0.67	0.96	8YSZ-1	14502 h	6.09	0.75	0.67	1.01
5YSZ-2	1400-2 h	3.39	0.74	0.70	0.96	8YSZ-2	1400-2 h	6.24	0.84	0.80	1.01
5YSZ-3	1500-1400-2 h	3.76	0.79	0.60	1.19	8YSZ-3	1450-1350-2 h	5.84	0.76	0.65	1.07
5YSZ-4	1450-1400-5 h	3.95	0.80	0.68	1.18	8YSZ-4	1400-1350-5 h	5.91	0.82	0.72	1.08
5YSZ-5	1450-1350-5 h	4.37	0.74	0.64	1.09	8YSZ-5	1400-1300-5 h	5.77	0.82	0.72	1.04
5YSZ-6	1450-1350-10 h	3.49	0.80	0.65	1.21	8YSZ-6	1400-1300-10 h	7.87	0.74	0.67	0.99
5YSZ-7	1400-1350-10 h	4.52	0.75	0.65	1.11	8YSZ-7	1350-1300-10 h	7.2	0.77	0.69	1.08

## Data Availability

Data are contained within the article.

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
