# Peer review of "Effect of Two-Step Sintering on the Mechanical and Electrical Properties of 5YSZ and 8YSZ Ceramics"

_materials, 2023, doi:10.3390/ma16052019_

Round 1

Reviewer 1 Report

In this paper, the density, average gain size, phase structure, mechanical and electrical properties of conventionally sintered and two-step sintered 5YSZ and 8YSZ are investigated in detail.

The article provides a detailed analysis of two-stage sintering of yttrium-stabilized ceramics. The data obtained in the article are of great importance for researchers and engineers involved in the production of ceramic products from this type of ceramics.

The article details the mechanisms of diffusion and conductivity in ceramics obtained by two-stage sintering depending on the temperatures T1 and T2. In terms of methodology, everything is done correctly.

The conclusions are consistent with the evidence and arguments presented and they address the main question posed.

The references are appropriate.

The tables are informative, the figures fully reflect the essence of the experiment.

Author Response

We gratefully thanks for the precious time the reviewer spent making constructive remarks. Thanks very much for taking the time to review this manual. Thank you very much for your recognition of this work!

Reviewer 2 Report

The manuscript by Li et al reports on the electrical and mechanical properties of a two samples prepared by the two-step sintering. Results are interesting and worth of publication. I just have some small comments for authors.

 Abstract. It is said “As the grain size of YSZ ceramics became smaller, dense YSZ materials with submicron grain size and low sintering temperature were optimized in terms of mechanical and electrical properties.” Please clarify this sentence. What do you mean with “as the grain size of YSZ ceramics became smaller”

Abstract. It is said “5YSZ and 8YSZ in TSS process significantly improved the plasticity, toughness, and electrical conductivity of the samples and significantly suppressed the rapid grain growth.” Please define TSS as it is the first time you are using it.

Introduction. Please define TSS.

Results. Table 1 is confusing. The time in the table corresponds to the first temperature, T1, or to the second, T2. Please be more specific and clarify it.

Reviewer 3 Report

The authors have studied the effect of two-step sintering on the mechanical and electrical properties of 5YSZ and 8YSZ ceramics in comparison with conventionally sintered ceramics. They found that the hardness of the samples is mainly affected by the relative density, and the fracture toughness and elastic modulus of the ceramic bodies are increased due to the grain refinement effect. The total conductivity of the samples was mainly influenced by grain size and grain boundary area. The research is well designed and presented clearly. A good comparative analysis of existing publications concerning the tasks set in the work is performed. The methodological section of the manuscript is presented in sufficient detail. The authors used the modern equipment for preparation and test of samples as well as visualization and assistance in the interpretation of the obtained results.

However, some shortcomings should be corrected to make the manuscript acceptable for publication in Materials.

(1) In my opinion, the title of the manuscript should be corrected, namely the word “ceramics” should be added in the end.

(2) In the last paragraph of the Introduction part, the authors should indicate the potential field of application of the studied materials.

(3) Sub-section 2.1: The spelling should be checked, namely the capital letter in the beginning of sentences etc. The phrase “TSS was raised to T1…” should be corrected as the subject and predicate should be “Temperature” and “was raised”, respectively.

(4) Sub-section 2.2: In the sentence “The fracture toughness KIC was determined by measuring the indentation in the following Equation (3): [25, recent references]”, a few recent references should be added to substantiate selection of Equation (3) for determination of fracture toughness of YSZ ceramics. The following references are suggested: https://doi.org/10.5604/01.3001.0015.2625, https://doi.org/10.3390/ma15082707. This will increase the weight and significance of the research.

(5) The labels (a) to (j) in the images of Figure 4 cannot be recognized. The font size should be increased. The same concerns the font in the scale bars.

(6) The caption to Figure 9 corresponds to 5YSZ ceramics whereas 8YSZ ceramics are given in legend in Figure 9(c). This discrepancy should be fixed.

(7) The caption to Figure 10 corresponds to 8YSZ ceramics whereas 5YSZ ceramics are given in legend in Figure 10(c). This discrepancy should be fixed.

Round 2

Reviewer 3 Report

All the reviewer’s comments were taken into account by the authors. The manuscript can now be accepted for publication in Materials.